# Between Promise and Proof: Evaluating PRP’s Role in Modern Gynecology

**DOI:** 10.3390/medicina61091514

**Published:** 2025-08-23

**Authors:** Andreea Borislavschi, Aida Petca

**Affiliations:** 1Department of Obstetrics and Gynecology, “Carol Davila” University of Medicine and Pharmacy, 8 Eroii Sanitari Blvd., 050474 Bucharest, Romania; andreea.borislavschi@drd.umfcd.ro; 2Department of Obstetrics and Gynecology, Elias University Emergency Hospital, 17 Mărăști Blvd., 050474 Bucharest, Romania

**Keywords:** platelet-rich plasma, PRP, gynecology, menopause, stress urinary incontinence

## Abstract

Autologous platelet-rich plasma (PRP) has emerged as a promising regenerative therapy in various medical fields, including the treatment of stress urinary incontinence (SUI) in women. PRP promotes regeneration by delivering a concentrated dose of platelets to damaged tissues, triggering healing mechanisms such as hemostasis, revascularization, and connective tissue regeneration through the release of growth factors. Despite evidence supporting the short- and medium-term benefits of PRP, its long-term efficacy remains unclear, largely due to the limited duration of follow-up in existing studies. Furthermore, the lack of standardized protocols for both preparation and administration of PRP poses a significant challenge to accurately assessing and comparing its sustained therapeutic outcomes. This literature review utilized comprehensive searches of PubMed and Google Scholar databases to analyze current evidence regarding PRP’s role in managing SUI in women. SUI, often resulting from weakened pubo-urethral ligament or intrinsic sphincter dysfunction due to childbirth, aging, or hormonal changes, significantly impacts quality of life. PRP offers a safe, minimally invasive, and cost-effective treatment option; however, further large-scale, well-designed studies are necessary to define optimal protocols and confirm long-term benefits. Advancing understanding of PRP therapy could substantially improve clinical management and patient quality of life in SUI.

## 1. Introduction

Autologous platelet-rich plasma [PRP] is a relatively novel treatment embracing more and more medical fields. PRP has been used for over 30 years for various indications, highlighting its considerable potential in regenerative medicine [1]. The main indication of PRP treatment is tissue regeneration [1].

The first use of PRP treatment was in 1977, as a tissue-sealing agent in dogs, to support stronger fibrin polymerization [2]. The sealant successfully stopped leakage of cerebrospinal fluid fistulas in dogs; in acute instances, treatment with grafts plus PRP showed success in six/six cases, whereas treatment with grafts alone was unsuccessful in zero/five cases [2]. Later on, its healing ability has been extensively valued and harvested across various medical fields, particularly in regenerative medicine [3]. The first therapeutically documented use of autologous PRP in humans was in 1987, in cardiac surgery, for intraoperative blood salvage to aid in healing and reduce bleeding [4].

PRP is the prepared liquid fraction of autologous peripheral blood that has concentrated platelets [5]. The underlying scientific hypothesis for concentrated platelets is that, when injected at the site of damaged tissue, they may initiate healing by triggering the hemostatic cascade, revascularization, and neoformation of connective tissue [1]. Its regenerative ability is primarily acquired by stimulating a supraphysiological release of growth factors that jump-starts the healing process [6]. 

The tissue healing cascade consists of many players that work synergistically [1]. This process involves a coagulation cascade (where platelets activate and release their granules’ contents), formation of the fibrin mesh (polymerization of fibrinogen), and platelet plug development (which acts as a temporary extracellular matrix, allowing cells to proliferate and differentiate) [5,7].

Despite encouraging preliminary outcomes, a significant gap in knowledge remains in the use of PRP as a treatment for women with stress urinary incontinence (SUI). While small-scale studies revealed short- to mid-term improvement in symptoms, critical aspects such as the optimal PRP preparation protocol, injection site, treatment frequency, and long-term efficacy remain under-investigated. Small cohort studies, a lack of standardized methodologies, and short-term follow-ups limit current literature. Additionally, there is no consensus on patient selection criteria or how comorbidities may affect outcomes. This lack of large-scale, randomized controlled trials hinders the integration of PRP into standardized clinical practice and underlines the urgent need for more rigorous research to validate its role in the management of women’s SUI [1,2,3,4,5,6,7].

This narrative review aims to provide a comprehensive overview of the current understanding and potential clinical application of PRP in the treatment of SUI in women. The review outlines significant studies found in the literature on this topic, the biological basis and mechanisms of action of PRP, discusses the various methods of PRP preparation, and contextualizes its relevance within the pathophysiology of SUI in women. By integrating findings from preclinical and clinical studies, this work highlights the rationale for PRP use in regenerative urology and identifies gaps in knowledge that warrant further investigation.

The chronic nature of the disease, coupled with its frequent association with aging and comorbid conditions, poses considerable challenges for both patients and healthcare providers. Current treatment modalities, ranging from conservative approaches to surgical interventions, often fail and may carry limitations related to efficacy, tolerability, or patient eligibility. The aim of this narrative review is to provide clinicians, especially early-career physicians, with a concise yet comprehensive synthesis of recent evidence on this topic. To support a deeper understanding of its therapeutic rationale, this review also includes a brief recap of the pathophysiological mechanisms underlying SUI.

## 2. Materials and Methods

This literature review identified relevant peer-reviewed studies on the use of PRP in women diagnosed with SUI. A comprehensive search was conducted using four major databases: PubMed, Web of Science, Embase, and Google Scholar, encompassing studies published up to July 2025, with emphasis on clinical trials and reviews addressing PRP use in women diagnosed with SUI. Keywords such as “platelet-rich plasma,” “PRP,” “stress urinary incontinence,” “female incontinence,” and “regenerative therapy in gynecology” were used in various combinations. Inclusion criteria encompassed clinical studies (randomized controlled trials, observational studies, and case series), reviews, and relevant experimental research published in English. Exclusion criteria were duplicates and articles lacking methodological clarity or outcome data, or addressed other types of urinary incontinence, animal studies, or were published in languages other than English (Figure 1). While the final reference list includes approximately 50 sources, not all are original studies; many serve to provide essential background information on PRP biology, preparation techniques, and the pathophysiology of female SUI. This approach was chosen to enhance the review’s educational value, particularly for early-career clinicians, and to support a more integrated understanding of this emerging therapeutic field.

## 3. Content of PRP

PRP treatment is designed to mimic the healing process in a local microenvironment [8]. The biology of platelet concentration is extremely complex [6]. PRP products are living biomaterials whose clinical outcomes are dependent on the patient’s blood [its intrinsic, versatile, and adaptive characteristics] and local microenvironment [6]. There are no clear protocols regarding the formulation and content of PRP; therefore, its composition can vary in terms of the number of red blood cells (RBCs), leukocytes, platelets, and growth factors [9,10,11].

PRP is sorted into three different groups: pure platelet-rich fibrin (P-PRF), leukocyte-rich PRP (LR-PRP), and leukocyte-poor PRP (LP-PRP) [12].

Leukocytes possess immune and host-defense mechanisms that influence chronic tissue lesions [1]. Lymphocytes produce insulin-growth factors, playing a positive role in tissue remodeling [1]. LR-PRP and LP-PRP have been a subject of debate among scientists [13]. An in vitro study suggested that LR-PRP has a rather catabolic effect on tissues, whereas LP-PRP has an anabolic effect [14], a finding also supported by an in vivo study, which reported an acute inflammatory response associated with LR-PRP [15]. The inflammatory response has a significant impact on the healing process; however, the variables that determine the positive or negative effects of PRP treatment are still being studied [13] (Table 1).

Monocytes are very important participants in this process, due to their plasticity and ability to differentiate into macrophages [13]. Macrophages have the ability to phagocytose cells that undergo apoptosis [13]. There are two phenotypes of macrophages, which have antagonistic actions: M1 pro-inflammatory (promoting host defense) and M2 anti-inflammatory (with a healing function) [13,16]. Macrophages promote angiogenesis by producing angiogenic factors and cytokines [13].

Platelets have roles that go far beyond hemostasis. They contain granules with various biomolecules relevant to hemostasis, inflammation, regeneration, and immune function [13]. Regarding their regenerative ability, the most important part of platelets is the abundant number of growth factors within their granules: vascular endothelial factor (VEGF), platelet-derived growth factor (PDGF), epidermal growth factor (EGF), hepatocyte growth factor (HGF), transforming growth factor (TGF), and fibroblastic growth factor (FGF) [13]. The platelets’ granules also contain cytokines, chemokines, and other microparticles, all of which play a key role in tissue regeneration [13]. The activation of platelets results in a change to their cytoskeleton, leading to the formation of pseudopods that spread across the injured zone (platelet aggregation) [6]. Through these pseudopods, the granular contents are released [6]. Thrombin, calcium chloride, or collagen represent essential factors that activate platelets; thrombin is the most potent [6].

The role of RBCs in tissue regeneration is unknown [1]. Due to the fact that RBC membranes disintegrate easily and their content has a cytotoxic effect on tissues, it is reasonable to assume that RBCs lower the quality of PRP [1]. RBCs from PRP injections cause an important pro-inflammatory response, leading to significant local cellular dysfunction [17]. Therefore, it is vital to limit the RBC contamination of the PRP [1].

## 4. Tendon and Ligament Healing

Tendon healing is a complex process where growth factors play a central role: insulin-like growth factor 1 (IGF1), TGF, VEGF, PDGF, and FGF [18]. Tendons heal spontaneously, but the scar tissue is mechanically inferior. The healing process is similar to that of other tissues. Upon tissue damage, blood vessels rupture, and the intrinsic cells initiate the coagulation cascade, forming a blood clot [18]. The platelets from the clot release multiple molecules, including growth factors, which trigger an acute inflammatory response [18]. Extrinsic cells [neutrophils and macrophages] phagocytose the debris. A second battery of cytokines is released by the intrinsic and extrinsic cells, marking the beginning of the regenerative phase [18]. This phase is represented by granulation tissue formation, neovascularization, and fibroblast proliferation (extracellular matrix synthesis); all lead to callus formation [18]. The last step consists of remodeling the callus by organizing and orienting the collagen within the healthy matrix outside the damaged tissue [18] (Figure 2).

Connective tissues, such as tendons, ligaments, and muscles, heal through three distinct phases: inflammation, proliferation, and remodeling. Cytokines play a pivotal role throughout this process by binding to specific transmembrane receptors, activating intracellular signaling cascades that modulate nuclear gene expression. The outcome represents the production of proteins that regulate cellular proliferation, chemotaxis, angiogenesis, differentiation, and extracellular matrix synthesis. The cytokines and growth factors released from PRP are known to modulate these fundamental biological processes, thereby facilitating tissue repair and regeneration within the musculoskeletal system [19].

PRP plays differing roles in tendon healing depending on whether the condition is an acute injury or chronic tendinosis, as the underlying biological processes vary significantly. In acute tendon injury, the injection of growth factors enhances tendon callus strength and stiffness when injected shortly after injury. It promotes recruitment of circulating progenitor cells and collagen synthesis at the injury site. In chronic tendinosis, PRP treatment demonstrated the ability to stimulate tenocyte proliferation, upregulate expression of collagen, and enhance extracellular matrix production. It is thus suggested that PRP exerts both regenerative and modulatory effects with therapeutic potential that varies according to the phase and nature of the tendon pathology [19].

SUI is a pathology often encountered in adult women, with a prevalence of 40% [20]. Aging, obesity, estrogen deprivation, and birth trauma are well-known risk factors [20]. In agreement with the integral theory, SUI’s etiopathogenesis is represented by the damage of the pubo-urethral ligament [21]. A variety of treatments are available, such as non-invasive (pelvic floor muscle exercises, changing the lifestyle, electrostimulation) and invasive methods (mid-urethral tapes and colposuspension) [20]. Nikolopoulos and colleagues have stated the plausibility of PRP treatment in SUI since 2016 [22]. 

Thus, it is fair to assume that a damaged ligament, such as the pubo-urethral ligament, is theoretically plausible to heal with the use of PRP injections.

The successful application of PRP in orthopedic medicine, particularly in the treatment of tendon and ligament injuries, has provided a solid foundation for exploring its regenerative potential in other connective tissues. In orthopedics, PRP has demonstrated the ability to enhance collagen synthesis, reduce inflammation, and accelerate the healing of chronically degenerated tendons and partially torn ligaments. These findings are of particular interest in the context of female SUI, where the integrity of the pubo-urethral ligament plays a central role in maintaining continence. Given the structural and functional similarities between musculoskeletal ligaments and the pelvic support system, the use of PRP as a regenerative therapy targeting the pubo-urethral ligament represents a biologically plausible and minimally invasive approach. This conceptual transition, from orthopedic applications to urogenital indications, forms the rationale for investigating PRP as a novel treatment strategy in women with SUI.

## 5. Preparation of PRP

Preparation protocols for the biologically active PRP cellular cocktail lack consensus regarding standardization [1]. Safe, easy, and ingenious preparation are the main advantages of PRP [23]. The employment of PRP biological treatments is complicated by the heterogeneity of PRP formulations and the poor standardization of evidence-based guidelines [1].

There are various PRP systems to facilitate the preparation of PRP that operate on a small volume of drawn blood and the principle of centrifugation [single or double spin] [24].

PRP is considered a blood component intended for non-transfusional use. Blood components are differentiated into two categories: for infiltrative/topical use or cellular therapy [25].

Following peripheral blood collection, the samples are subjected to centrifugation to separate the blood components. Variations in centrifugation time and G-forces result in significant differences in the yields, concentration, purity, viability, and activation status of the platelets [1,24,26]. It is reasonable to assume that a higher level of platelets should lead to a more positive outcome, as they release a higher level of growth factors. However, the correlation between the level of platelets and the efficiency of PRP does not align, and it cannot be controlled due to marked individual differences in baseline platelet counts [27]. Nguyen and Pham [28] stated that a very high concentration of growth factors may be counterproductive; the quantity of cell receptors and the downgrading effect of receptors should be considered [1].

A key factor in PRP efficiency is maximizing platelet purity, recovery, and outcome [27]. One study assessed various factors influencing PRP preparation and concluded that using tubes preloaded with anticoagulant, combined with a two-step centrifugation process, results in higher-quality PRP: first centrifuge at 300× *g* for 5 min, at 18 °C, followed by a second centrifuge [only the upper fraction] at 700× *g* for 17 min at 18 °C [27].

Thirty-nine studies on PRP preparation were reviewed, which showed great variability related to methods employed in different stages of PRP processing [29].

Eren et al. [30] investigated the impact of centrifugation duration and found that a single spin at 2600 rpm for 12 min yielded a higher VEGF concentration compared to a 10 min spin at the same speed. In a separate study, Yin et al. [31] examined various durations and centrifugal forces in a double-spin protocol, concluding that an initial centrifugation at 160× *g* for 10 min followed by a second spin at 250× *g* for 15 min produced the highest-quality PRP formulation [29].

Single centrifugation vs. double centrifugation has also been evaluated [29]. Carofino et al. [32] found that single centrifugation produced a marginally higher platelet concentration than double centrifugation (3.6 vs. 3.4). Single centrifugation was performed at speeds below 1500 rpm for 5 min, while the double centrifugation protocol included a second spin at speeds under 6300 rpm for 20 min [32].

Another study which evaluated platelet concentration compared three protocols: the first protocol involved a single centrifugation under 500 rpm for 5 min, the second protocol consisted of a single centrifugation under 3200 rpm for 15 min, and the third protocol employed a double centrifugation with an initial spin at under 1500 rpm for 5 min followed by a second spin at under 6300 rpm for 20 min [29]. The results stated that protocol 2 reached a higher platelet concentration [29].

Kutlu et al. [33] compared three protocols for PRP preparation: protocol 1 (single centrifugation at 1000 rpm for 10 min), protocol 2 (double centrifugation: first spin at 2400 rpm for 10 min and a second one at 3600 rpm for 15 min), and protocol 3 (double centrifugation: first spin at 3000 rpm for 3 min and another one at 3000 rpm for 13 min). Further analysis revealed that protocols 2 and 3 have no significant difference between them and have a higher platelet concentration than protocol 1 [29,33].

Carofino et al. [32] evaluated the effect of PRP on tenocytes proliferation, detecting that cell viability is slightly decreased in the presence of anesthetics (lidocaine) and corticosteroids [32].

Overall, effective PRP has been characterized as a complex balance of autologous multicellular components in a small amount of plasma that is collected from a fraction of peripheral blood after centrifugation [1].

What are the necessary characteristics of a PRP sample to maximize its regenerative potential? This important question requires further investigation, and its answer may pave the way for standardized protocols that enhance sample quality and advance our understanding of regenerative medicine [1].

Several factors can affect the clinical efficacy of PRP samples, including patient age, use of antiplatelet therapy, bone marrow aplasia, uncontrolled diabetes, sepsis, cancer, as well as the quality and quantity of platelets and the PRP preparation method [25,34].

Research on the optimal platelet concentration for cell regeneration in vitro reveals considerable variation and has yet to identify a conclusive ideal level [25]. An overall trait is observed: a relatively high platelet concentration is best suited to mimic the in vivo situation; lower or hyper-concentrations may be inappropriate [25]. The author suggests a platelet concentration of 1.0–1.5 × 10^6^/μL and the optimal PRP/media ratio to be PRP < 10% (Vol/Vol) [25] (Table 2).

There is currently no universally standardized protocol for its preparation. PRP can be prepared through manual or semi-automated methods. Among the most commonly used systems in gynecologic practice are commercial kits, which offer more reproducible results and ease of use compared to manual techniques.

## 6. Clinical Use of PRP in Stress Urinary Incontinence (SUI)

In a prospective pilot study, Cheng-Yu et al. [20] reported the effectiveness of autologous PRP injections in women diagnosed with SUI. A total of 20 women were administered a one-time PRP injection in the anterior vaginal wall (mid-urethral) [20]. PRP was prepared using a commercial kit, following its standardized procedures: two tubes preloaded with anticoagulant were centrifuged at 3400 rpm for 15 min [20].

The efficiency of the treatment was assessed with the help of questionnaires (ICIQ-SF International Consultation on Incontinence Questionnaire-Short Form, UDI-6 Urogenital Distress Inventory, IIQ-7 Incontinence Impact Questionnaire, OABSS Overactive Bladder Symptom Scores, POPDI-6 Pelvic Organ Prolapse Distress Inventory 6) [20]: pre-treatment, at 1 month post-treatment, and 6 months post-treatment. Regarding the type of SUI [20],

Mild SUI—1 patient; status—improved [20];Moderate SUI—12 patients; status—6 improved and 6 unchanged/worse [50%] [20];Severe SUI—7 patients; status—5 improved and 2 unchanged/worse [71.4%] [20].

One-time PRP injection detects a satisfactory response both at 1 month and 6 months post-treatment in SUI [20].

Another study about women with SUI highlights the possibility of PRP treatment as an effective alternative outpatient procedure [36]. A total of 20 women were enrolled in the study and attended all follow-ups. They underwent two PRP treatments consisting of two PRP injections in the lower third of the anterior vaginal wall at 4- to 6-week intervals [36]. A significant improvement in symptoms was observed at 3 months post-treatment, with further progress at 6 months post-treatment (80% of women reported to be at least improved) [36].

Contemporary literature weighs the effectiveness of this treatment.

A recent clinical study involving only 50 patients aimed to answer an interesting question: Is it a placebo effect, or is it actually working? The main flaw in this randomized, placebo-controlled trial is that they evaluated the effectiveness of a single PRP injection into the anterior vaginal wall at the mid-urethra. Both groups showed comparable outcomes on objective measures, and the safety profiles were similarly favorable, with only minor adverse events reported. A single administration may be insufficient to produce meaningful clinical improvement. Further research is warranted due to the contradictory results of these two studies regarding a single injection of PRP in women with SUI [20,37]. The variable outcomes of single-injection PRP may reflect placebo effects or subtherapeutic dosing. Given the regenerative nature of PRP, repeated treatments are likely needed to achieve consistent and sustained efficacy.

There are multiple treatment options available for postmenopausal women with stress urinary incontinence. A parallel has been drawn between PRP treatment and intravaginal phytoestrogen gel. A prospective randomized clinical study has shown that repeated autologous PRP injections at the mid-urethral site are more effective than intravaginal phytoestrogen gel in treating SUI in postmenopausal women. After six months, both treatment groups showed significant improvements from baseline in symptom severity and quality-of-life measures; however, the PRP group exhibited significantly greater reductions, along with a higher overall success rate. The PRP is prepared by using 1.5 cm of anticoagulant drawn into a 20 cc syringe, followed by 14 cm of peripheral venous blood from the patient, and transferred into a specialized PRP tube. Centrifugation was performed using a computerized device at either 4000 RPM for 10 min or 3500 RPM for 5 min to separate the blood components. The PRP layer (the white part) was elevated above the line in the narrow section of the tube using the built-in control. A syringe was then used to withdraw 2 cm of PRP from the bottom of the tube. The sites of injections using 8 mm of PRP were into the anterior vaginal wall at three sites on each side near the external urethral sphincter. The injection targeted the paraurethral region, extending approximately 10 mm between the external urethra and the lateral vaginal wall [38] (Table 3).

A double-blind, randomized, sham-controlled trial demonstrated that periurethral injection of PRP is a safe and efficacious intervention for the treatment of SUI in women. Participants who received PRP showed a statistically significant improvement in subjective symptom scores, a higher rate of self-reported cure, and a greater reduction in objective measures of urine loss compared to those in the sham group. No adverse events were reported, underscoring the favorable safety profile of this procedure. This study concludes that PRP treatment is a potential minimally invasive and well-tolerated therapeutic option for women with SUI [39].

A recurring limitation in studies investigating PRP for stress urinary incontinence is the lack of detailed reporting on preparation protocols. Essential technical aspects (centrifugation steps, rotational speed, processing time, and the use of activators) are frequently omitted or vaguely described. References to “standard protocols” or commercial kits often lack methodological clarity, reducing reproducibility and hindering cross-study comparisons. This underscores the need for standardized reporting to support consistent and evidence-based application of PRP in gynecologic practice.

## 7. Intrinsic Sphincter Deficiency (ISD)

A challenging problem to address in clinical practice is intrinsic sphincter deficiency (ISD). Historically, various approaches have been tested, but none have proven satisfactory. These treatments included the first-line conservative non-invasive methods, such as behavioral modifications, pad use, pelvic floor exercises, and pharmacotherapy [34]. The second-line treatment involved invasive procedures such as midurethral slings and bulking agents; females with mild SUI might hesitate to choose surgery, all the more since the FDA warned about the complications of slings [34]. Nowadays, minimally invasive treatments in this distressing disorder have been developed: injection therapy with urethral bulking agents, fat graft, or stem cell formulations [25,34].

Regarding SUI due to intrinsic sphincter deficiency, there is evidence to support PRP efficiency [34]. Twenty-six women diagnosed with SUI due to ISD were enrolled in a study and treated with four repeated mid-urethral PRP monthly injections [34]. Approximately eighty percent (80.8%) reported a positive response and a decrease in symptom severity [34]. After peripheral blood collection, the preparation of PRP included a two-centrifugation step: the first one at 200× *g* for 20 min at 20 °C, and a second one [only with the upper layer] at 2000× *g* for 20 min at 20 °C [32]. Fifty percent of women reported a successful outcome, 46.2% were completely dry at 3 months, and 26.9% maintained continence 1 year after [34]. 

Literature search highlighted another study regarding men with SUI due to ISD, proving the effectiveness of PRP injections [40]. Thirty-five male patients were included in the study; 20% achieved complete dryness, and 40% experienced moderate improvement. The preparation of PRP is not specified. Yuang-Hong et al. concluded that PRP treatment could be an alternative treatment for both females and males with moderate non-neurogenic SUI [40] (Table 4).

## 8. Why Use PRP Injections? PRP Treatment—To Whom It “May Concern”?

A relevant fact is that minor adverse reactions have been reported [20,34,36,40]. The safety component of PRP injections should not be underestimated. Other points in favor of PRP are not to be forgotten: it is easily produced, autologous, and cost-effective [20].

PRP can be easily produced in any hospital or outpatient care facility that owns a centrifuge machine. As it is an autologous treatment with a straightforward preparation, physicians should fully leverage its therapeutic potential.

A study concluded that the incidence of adverse reactions after intra-articular PRP treatment for knee osteoarthritis was interestingly related to leukocyte concentration, such as pain and swelling [41]. From a theoretical perspective, leukocyte-rich PRP may induce a pro-inflammatory cascade due to its catabolic activity and the release of inflammatory markers [41]. However, evidence also suggests that LR-PRP may contribute positively to tissue remodeling, and the assumption of a purely catabolic effect is not universally supported. Its biological impact appears to be tissue-dependent [1,42]. Riboh et al. [42] stated that these adverse reactions appear to be a class effect of PRP, specifically related to leukocyte concentration [41]. Overall, regardless of leukocyte concentration, PRP treatment in knee osteoarthritis showed improvement in terms of pain and function [41]. In treating SUI with PRP injections, only one side effect was mentioned: one patient reported straining to void, which was self-limited [34]. Other side effects of PRP treatment are not described in the literature [1].

PRP was first used in sports medicine (orthopedics) in the early 2000s, reporting significant pain reduction in patients with chronic tennis elbow following buffered PRP injections [43]. In gynecology in 2016, PRP was mentioned in the form of a hypothesis suggesting its potential to restore the pubo-urethral ligament in stress urinary incontinence [22], and also a feasibility study in 2009 evaluated autologous platelet gel in pelvic organ prolapse surgery—but that was a preliminary, case-series approach, not a controlled clinical trial [44]. The actual use of it occurred in 2012 in treating cervical ectopy, with promising results [45].

Literature research has identified that most clinical evaluations of the therapeutic use of PRP for SUI in women have been conducted on relatively small cohorts, typically ranging from 20 to 50 participants per study group. The limited sample sizes in current studies significantly restrict the statistical power of their findings and limit the applicability of the results to the broader patient population. While many of these studies report favorable outcomes, the small cohorts increase the likelihood of statistical error and reduce sensitivity to detect modest yet clinically relevant differences between PRP and control or placebo interventions. From both statistical and clinical perspectives, small-scale trials weaken the reliability and validity of the conclusions that can be drawn. Thus, these studies are more vulnerable to methodological biases, including selection and reporting biases, and frequently fall short of the robustness seen in large-scale, multicenter, double-blind randomized controlled trials. Consequently, despite promising preliminary evidence supporting the use of PRP in SUI, current data remain insufficient to support its routine clinical application. There is a critical need for larger, well-designed trials employing standardized methodologies to substantiate efficacy, define optimal treatment parameters, and assess long-term safety [46].

Regarding SUI in women, the benefits of PRP treatment are clearly stated in literature [20,34,36,40]. Whereas in women with pelvic organ prolapse [multiple compartment damage], such as defect of the central and/or posterior compartment, the benefits are unknown and need to be further explored [47].

In order to draw attention to PRP treatment for SUI in women, many studies have shown beneficial traits [25]. However, it is essential to acknowledge the existing knowledge gaps, specifically regarding optimal preparation methods, injection sites, and the frequency of PRP administration.

PRP treatment has often faced skepticism due to its growing popularity in private practices, raising concerns about its commercialization and potential exploitation by profit-driven providers. The lack of standardized protocols and regulatory oversight makes PRP vulnerable to being marketed as a panacea without sufficient evidence. In the gynecological area, SUI represents a condition that significantly and directly impairs patients’ quality of life. The use of PRP treatment is supported by well-designed clinical trials, including randomized, placebo-controlled, and sham-controlled studies, which demonstrate its safety and marked efficacy in alleviating symptoms and enhancing the quality of life. A key challenge that remains is the accurate diagnosis of the specific type of urinary incontinence to ensure the most effective treatment approach [48].

## 9. PRP Therapy: Lifelong Fix or Temporary Addiction?

The available data on following up the long-lasting effects of PRP treatment in SUI on women are very limited. Most studies evaluated up to 1 year. A notable study assessed the combination of PRP treatment with fractional CO_2_ laser treatment and followed up for up to 24 months, yielding promising results. Sixty-two women were enrolled in this study, and 62% of patients reported improved SUI symptoms [*p* < 0.001] at 12–24 months [49].

A study involving 26 women with ISD assessed the effects of four PRP treatments for SUI. After treatment, 80.8% reported improvement, with 50% achieving a successful outcome and 30.8% experiencing a mild benefit. The success rate was higher in patients with no prior surgery (58.3%) compared to those with previous sling procedures (42.9%), though the difference was not statistically significant. The mean follow-up was 12.6 months (range 12–26). Five women (19.2%) saw no improvement, most of whom had complicating conditions such as mixed incontinence or neurologic disease. The treatment effect was generally sustained over 12 months, with only a few reporting symptom recurrences [34].

Current literature supports PRP treatment for SUI in women, showing its safety and promising short- to mid-term efficacy. These effects persisted for up to 12 months without significant symptom rebound, suggesting a stable therapeutic response in many patients. However, a notable constraint in the current literature is the lack of long-term follow-up data. Most studies monitor outcomes only up to one year, with very few extending to 15 or 24 months, and almost none evaluating durability beyond that. This significant drawback raises a crucial concern: Is PRP a permanent regenerative treatment, or does it become a chronic therapy requiring repeated sessions? Without standardized protocols and large, multicenter randomized controlled trials with long-term follow-up, the sustainability of PRP’s therapeutic effect remains unclear; therefore, it should be considered investigational. Clinicians must counsel patients accordingly, emphasizing that long-term efficacy and treatment frequency are still under investigation; further high-quality studies are urgently needed to clarify these aspects [34,46,49].

## 10. Targeting the Hammock: Exploring PRP Injection Sites in SUI

Urinary continence in women is sustained through a coordinated interaction between anatomical integrity and neuromuscular function. This includes the integrity of the urethral sphincter, sufficient support from pelvic floor muscles and connective tissues (such as the endopelvic fascia and pubo-urethral ligaments), and proper coordination with bladder function. According to the hammock theory (integral theory), supportive elements create a suburethral backboard that stabilizes and compresses the urethra during elevations in intra-abdominal pressure (coughing, sneezing, exercise), thereby preventing involuntary urine leakage [21,50].

Urinary incontinence, particularly SUI, occurs when this supportive system is weakened or damaged. Specifically, the failure of the urethra to maintain closure during episodes of increased intra-abdominal pressure results in involuntary urine leakage. The pathophysiology of SUI typically involves anatomical deficiencies (such as urethral hypermobility and damaged pelvic fascial support), as well as ISD. Consequently, therapeutic interventions are designed to restore both structural support and sphincteric competence [21,50,51].

The hammock theory suggests that augmenting suburethral support can restore continence, guiding the targeted use of PRP [50]. Accordingly, various studies have administered PRP directly into the periurethral tissues or the urethral sphincter to harness its regenerative potential [34,40,52]. Chiang et al. injected 1 mL of PRP at five positions (2, 5, 7, 10, and 12 o’clock) around the urethral meatus in women with ISD, achieving sustained symptom relief for up to 12 months post-treatment [34]. Similarly, Jiang et al. injected 5 mL of PRP circumferentially into the external sphincter at five sites under cystoscopic guidance (a location closely corresponding to the urethral support hammock), reporting significant increases in abdominal leak-point pressure and improvement in visual analogue scores (VASs) [40]. Periurethral PRP injections have also been described by Clavijo et al. [52], with injections at the 3, 6, 9, and 12 o’clock positions, reinforcing external compression around the urethra to improve continence [53].

Although specific protocols differ slightly across studies, they consistently target the anatomical “hammock” region for PRP administration. The objective is to stimulate tissue regeneration, promote collagen synthesis, and reinforce urethral support through minimally invasive, regenerative techniques [34,40,51,52]. Further comprehensive research is required to determine whether injection site variations significantly influence clinical outcomes.

## 11. Patient-Specific Determinants of Platelets in PRP

The cellular composition and therapeutic potential of PRP are significantly influenced by individual patient characteristics. Studies have demonstrated a significant inverse relationship between age and platelet concentration, with an estimated decline of approximately 32,000 platelets/μL per decade, impacting both whole blood and PRP preparations [53]. In contrast, baseline platelet count has consistently emerged as the strongest predictor of platelet yield in PRP, with reports indicating up to a 3.8-fold increase in final PRP concentration relative to initial whole blood values [54]. While body mass index (BMI) has been hypothesized to affect PRP composition, recent evidence suggests no statistically significant correlation between BMI and platelet count in PRP. However, some data indicate that BMI may subtly influence the relative concentration of platelets during centrifugation, though its clinical relevance remains uncertain [54]. Overall, age and baseline hematologic parameters appear to be the primary determinants of PRP quality.

## 12. Limitations

While literature reviews serve as valuable tools for synthesizing existing research on PRP treatment for SUI in women, they are subject to several limitations. The available studies vary widely in their design, sample sizes, follow-up periods, and outcome measures, which makes it difficult to compare results and limits the ability to apply conclusions broadly. Selection bias may result from language restrictions, limited access to full-text articles, or the search strategies employed during the review process. Additionally, publication bias—where studies with positive findings are more likely to be published—can distort the overall assessment of PRP efficacy. Unlike systematic reviews and meta-analyses, narrative literature reviews often lack rigorous methodological frameworks and quantitative synthesis, thereby limiting their capacity to provide definitive and robust conclusions.

Moreover, there is significant heterogeneity in PRP preparation techniques, with many studies failing to report key technical parameters such as centrifugation speed and time, platelet concentration, leukocyte content, or activation method. This lack of standardization complicates cross-study comparisons and undermines reproducibility.

## 13. Future Directions

Future research on PRP therapy for women with SUI should prioritize standardization and clinical validation. Future trials should adopt uniform classification systems for PRP preparation, composition, and injection site. Establishing more stringent patient selection criteria, accounting for factors such as age, baseline platelet count, and relevant comorbidities (including diabetes, obesity, or hormonal status), is essential to assess and optimize the therapeutic efficacy of PRP accurately, as these variables may significantly influence tissue responsiveness and regenerative outcomes. The methodological shortcomings highlight the need for future high-quality randomized controlled trials with standardized PRP protocols, adequate sample sizes, and consistent, validated outcome reporting to clarify the true efficacy and clinical value of PRP in the management of SUI. Furthermore, given the growing body of heterogeneous evidence, a comprehensive systematic review and meta-analysis would be instrumental in synthesizing available data, identifying knowledge gaps, and guiding future research and clinical practice.

Future studies should incorporate extended follow-up and standardized outcome measures to better evaluate the durability and clinical utility of PRP in the management of women with SUI. At present, proposing a standardized PRP preparation framework in gynecology is premature, as few studies report sufficient methodological detail to support evidence-based recommendations.

The interpretation of PRP outcomes in SUI is influenced by the potential for placebo effects, particularly in studies without proper blinding or control groups. Since symptom improvement is often self-reported, perceived benefit may reflect expectation rather than objective change. Additionally, the literature shows substantial inconsistency in PRP preparation, injection techniques, and outcome measures, making direct comparisons difficult. These methodological differences underscore the importance of standardized protocols and consistent reporting in future studies. Nevertheless, the majority of published studies report clinically meaningful symptom improvement, suggesting a therapeutic signal that warrants further investigation through standardized, controlled research.

This raises the important question of whether PRP offers a true regenerative cure or merely provides temporary symptomatic relief requiring repeated administration. The limited availability of long-term follow-up data, combined with reports of symptom recurrence, raises the possibility that PRP acts more as a transient therapeutic enhancer.

## 14. Conclusions

As a treatment, PRP is extensively praised from a theoretical point of view, as seen in various medical fields, such as orthopedics, dermatology, and, more recently, in gynecology. PRP has shown promising effects on damaged connective tissue, both tendons and ligaments. A noteworthy characteristic in all medical fields is that the effects of PRP treatment tend to diminish over time, requiring the procedure to be repeated.

PRP is a novel, cost-effective treatment with proven benefits regarding SUI in women. Based on current literature, it is evident that protocols for PRP preparation require further investigation and standardization. Future studies should focus on enrolling larger patient cohorts, clearly defining optimal injection sites, and evaluating long-term efficacy to establish PRP as a reliable treatment for female SUI. A proper strategy for the treatment of SUI will lead to an increase in the quality of life, more so when such a treatment is minimally invasive and has no serious adverse reactions.

## Figures and Tables

**Figure 1 medicina-61-01514-f001:**
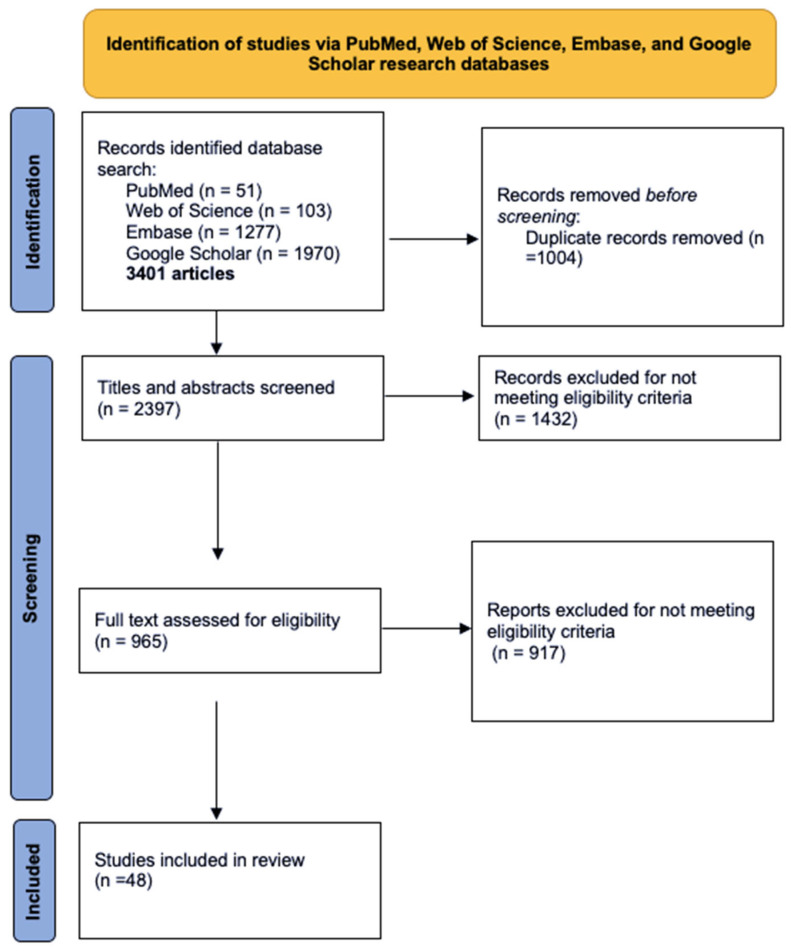
Search strategy in study selection process.

**Figure 2 medicina-61-01514-f002:**
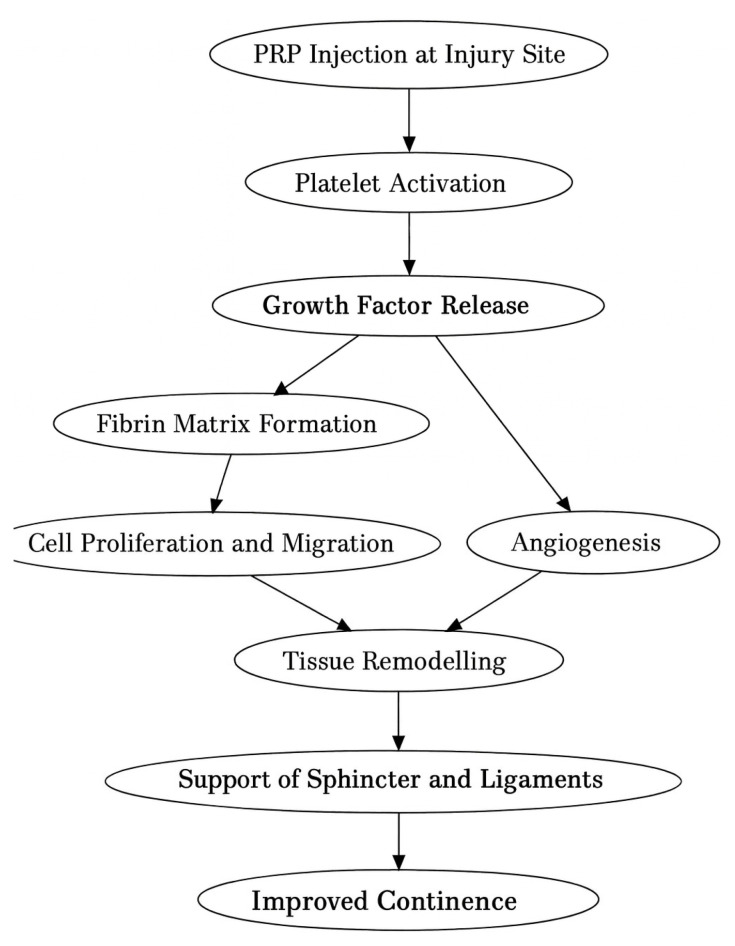
Mechanism of PRP treatment in SUI (stress urinary incontinence).

**Table 1 medicina-61-01514-t001:** PRP classification.

PRP Type/Component	Biological Role/Function	Key Notes/Classification
P-PRF	Fibrin-rich, platelet-dense matrix	Lacks leukocytes; used in slow-release regenerative applications
LR-PRP	Pro-inflammatory, immune activation	Contains leukocytes; may trigger stronger inflammatory responses
LP-PRP	Regenerative, less inflammatory	Reduced leukocyte content; associated with anabolic, tissue-repairing effects
Leukocytes	Host defense, inflammation	Present in LR-PRP; influence chronic tissue lesions
Lymphocytes	Tissue remodeling	Produce IGFs; contribute to regeneration
Monocytes	Differentiate into macrophages	Participate in inflammation resolution and healing
Macrophages (M1/M2)	Immune regulation, angiogenesis	M1: pro-inflammatory; M2: anti-inflammatory, healing phenotype
Platelets	Hemostasis, tissue repair	Contain granules rich in growth factors and cytokines
Growth Factors (e.g., VEGF, PDGF, EGF, HGF, TGF, FGF)	Stimulate cell proliferation and angiogenesis	Key mediators of PRP’s regenerative effect

PRP: platelet-rich plasma; PRP-F: pure platelet-rich fibrin; LR-PRP: leukocyte-rich PRP; LP-PRP: leukocyte-poor PRP; IGFs: insulin-like growth factors; VEGF: vascular endothelial growth factor; PDGF: platelet-derived growth factor; EGF: epidermal growth factor; HGF: hepatocyte growth factor; TGF: transforming growth factor; FGF: fibroblast growth factor.

**Table 2 medicina-61-01514-t002:** Various studies’ protocols for preparation of PRP.

Authors	Centrifuge Speed	Temperature	Number of Spins	Centrifugation Time
Amable et al. [27]	300× *g*; 700× *g*	18 °C	2	5 min; 17 min
Eren et al. [30]	2600 rpm	Not specified	1	12 min
Yin et al. [31]	160× *g*; 250× *g*	Not specified	2	10 min; 15 min
Carofino et al. [32]	<1500 rpm	Not specified	1	5 min
Carofino et al. [32]	<1500 rpm; <6300 rpm	Not specified	2	5 min; 20 min
Mazzocca et al.—protocol 1 [35]	<500 rpm	Not specified	1	5 min
Mazzocca et al.—protocol 2 [35]	<3200 rpm	Not specified	1	15 min
Mazzocca et al.—protocol 3 [35]	<1500 rpm; <6300 rpm	Not specified	2	5 min; 20 min
Kutlu et al.—protocol 1 [33]	1000 rpm	Not specified	1	10 min
Kutlu et al.—protocol 2 [33]	2400 rpm; 3600 rpm	Not specified	2	10 min; 15 min
Kutlu et al.—protocol 3 [33]	3000 rpm; 3000 rpm	Not specified	2	3 min; 13 min

rpm: rotations per minute; min: minutes.

**Table 3 medicina-61-01514-t003:** Studies revealing PRP preparation for treatment of SUI in women.

Author	Study Type	PRP Preparation Method	Centrifugation Details	Kit Used	Sites of Injection	Efficiency
Cheng-Yu et al. [20]	Prospective randomized clinical trial	Commercial kit, prepared using the manufacturer’s standardized protocol	3400 RPM for 15 min	Commercial kit unnamed	One single injection in the anterior vaginal wall	Mild SUI (1 patient) improved; Moderate SUI (12 patients) 6 improved and 6 unchanged/worse (50%); Severe SUI (7 patients) 5 improved and 2 unchanged/worse (71.4%)
Bedir [38]	Prospective randomized clinical trial	Centrifugation using a computerized device to separate blood components	4000 RPM for 10 min or 3500 RPM for 5 min	Unknown	Anterior vaginal wall at three bilateral sites, targeting the paraurethral region approximately 10 mm between the external urethra and lateral vaginal wall	PRP treatment exhibited significantly greater reductions, along with a higher overall success rate than intravaginal phytoestrogen gel
Athanasiou S [36]	Prospective randomized clinical trial	Unknown	Unknown	Unknown	Two injections in the lower one-third of the anterior vaginal wall at 4- to 6-week intervals	An amount of 80% of women reported to be at least improved
Ashton L [37]	Randomized placebo-controlled trial	Unknown	Unknown	Unknown	Anterior vaginal wall at the mid-urethra	Unable to demonstrate a difference in SUI treatment success between PRP and saline injections
Grigoriadis et al. [39]	Single-center, double-blind, randomized sham-controlled trial	Unknown	Unknown	Unknown	PRP injections at three levels of the urethra at 4- to 6-week intervals	PRP group demonstrated significantly higher rates of subjective cure, along with a notable reduction in urine loss on the 1 h pad test compared to the sham group at 6-month follow-up

SUI: stress urinary incontinence; PRP: platelet-rich plasma; RPM: rotations per minute.

**Table 4 medicina-61-01514-t004:** Studies on PRP in ISD.

Study	Patients	PRP Preparation Method	Injection Protocol	Clinical Outcomes
Chiang et al. [34]	26 women	Two-step centrifugation: 200× *g* for 20 min (20 °C), then 2000× *g* for 20 min (upper layer only)	Five sites around urethral meatus (2, 5, 7, 10, 12 o’clock); 1 mL PRP/site	A total of 80.8% reported symptom improvement
Yuang-Hong et al. [40]	35 patients (women and men)	Not specified	Five sites around urethral meatus (women); 1 mL PRP/site	A total of 20% reported complete dryness; 40% moderate improvement

PRP: platelet-rich plasma.

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
