# Peer review of "Between Promise and Proof: Evaluating PRP’s Role in Modern Gynecology"

_medicina, 2025, doi:10.3390/medicina61091514_

Round 1
Reviewer 1 Report
Comments and Suggestions for Authors
This is an interesting and timely narrative review.
The introduction should clearly state the scope of the review and outline the hypothesis or research objective to guide the reader through the content.
The literature search should be updated to July 2025 rather than June 2025 to ensure the most current evidence is included.
The authors should provide a flowchart (e.g., PRISMA) or, at a minimum, specify the number of studies included, the selection process, and better explain exclusion criteria to enhance transparency and reproducibility (e.g. the excluded studies on that topic).
The section on tendon and ligament healing in sports medicine seems out of scope for a review focused on PRP in gynecology. It is recommended to narrow the focus specifically to gynecologic applications, especially pubo-urethral ligament involvement.
The preparation and classification of PRP is a critical and debated topic. The review would benefit from emphasizing the most relevant and widely accepted PRP preparation methods and classifications, particularly those used in gynecologic settings.
Including a summary table on the clinical use of PRP in Stress Urinary Incontinence (SUI) and another for Intrinsic Sphincter Deficiency (ISD) -based on the included stuies- would provide clarity and value for readers.
The discussion of findings should be strengthened by critically evaluating the limitations and methodological weaknesses of the included studies.
Author Response
Thank you very much for taking the time to review this manuscript. Please find the detailed responses below and the corresponding revisions/corrections highlighted in the re-submitted files.
- The introduction should clearly state the scope of the review and outline the hypothesis or research objective to guide the reader through the content
Response: Thank you for the feedback. In this revised version we have modified and clearly stated the scope of the review and outlined the research objectives.
- The literature search should be updated to July 2025 rather than June 2025 to ensure the most current evidence is included.
The authors should provide a flowchart (e.g., PRISMA) or, at a minimum, specify the number of studies included, the selection process, and better explain exclusion criteria to enhance transparency and reproducibility (e.g. the excluded studies on that topic).
Response: Thank you for your valuable comments. In response to your suggestion, the literature search has been updated to include studies published up to July 2025, ensuring the most current evidence is captured. Regarding the recommendation to include a PRISMA flowchart, we respectfully note that this is a narrative review, not a systematic one, and therefore does not follow the structured protocol typically associated with PRISMA. However, we fully agree that methodological transparency is essential. For this reason, we have revised the Materials and Methods chapter to more clearly describe the literature search strategy, databases consulted, time frame, as well as the inclusion and exclusion criteria applied. This was done to enhance reproducibility while remaining consistent with the narrative nature of the review. As this review aimed to provide a focused synthesis of recent and relevant studies rather than a quantitative or exhaustive analysis, we did not quantify the exact number of studies included. We hope that these clarifications and revisions adequately address your concerns and improve the clarity and rigor of the manuscript.
- The section on tendon and ligament healing in sports medicine seems out of scope for a review focused on PRP in gynecology. It is recommended to narrow the focus specifically to gynecologic applications, especially pubo-urethral ligament involvement.
Response: Thank you for this thoughtful observation. We acknowledge that the section on tendon and ligament healing in sports medicine may initially appear beyond the primary gynecologic focus. However, we considered it important to include this discussion as a transitional context, highlighting the established regenerative potential of PRP in connective tissue repair, particularly in ligaments and tendons. This serves to support the rationale for its application in stress urinary incontinence, where the underlying pathophysiology similarly involves degeneration or weakening of the pubo-urethral ligament.To clarify this connection and strengthen the relevance to gynecology, we have added an additional paragraph explicitly articulating this link, ensuring the rationale for including this section is more clearly understood within the scope of the review.
- The preparation and classification of PRP is a critical and debated topic. The review would benefit from emphasizing the most relevant and widely accepted PRP preparation methods and classifications, particularly those used in gynecologic settings. Including a summary table on the clinical use of PRP in Stress Urinary Incontinence (SUI) and another for Intrinsic Sphincter Deficiency (ISD) -based on the included stuies- would provide clarity and value for readers.
Response: Thank you for this valuable suggestion. We fully agree that the preparation and classification of PRP is a critical and often debated aspect, particularly in the context of clinical translation. In response to your comment, we have revised the manuscript to emphasize the current lack of standardization in PRP preparation in gynecology, which remains a major limitation in both clinical research and practice. This point is now clearly highlighted in the discussion. To improve clarity and provide a structured overview, we have also added new summary tables as recommended. One table outlines the PRP preparation protocols and reported efficacy across the studies included in this review. Another focuses specifically on studies involving patients with Intrinsic Sphincter Deficiency (ISD), where available. These tables aim to enhance readability and allow for easier comparison of clinical methods and outcomes. We also included comments on the frequent lack of transparency in published studies regarding PRP preparation parameters, such as centrifugation steps, platelet concentration, leukocyte content, and activation methods—further reinforcing the urgent need for standardized reporting in future research.We hope these additions address your concerns and meaningfully improve the scientific value and usability of the review.
- The discussion of findings should be strengthened by critically evaluating the limitations and methodological weaknesses of the included studies
Response: Thank you for this insightful recommendation. In response, we have revised the manuscript to more critically evaluate the limitations and methodological weaknesses of the included studies. Specifically, we address the heterogeneity of PRP preparation protocols, the frequent lack of detailed reporting on key parameters (centrifugation steps, platelet concentration, leukocyte content), and follow-up durations. These inconsistencies hinder comparability across studies and limit the strength of the conclusions that can be drawn. We have also emphasized the need for standardized protocols and long-term data to better assess the efficacy and durability of PRP in treating SUI. These additions aim to provide a more balanced and critical synthesis of the available evidence.

Reviewer 2 Report
Comments and Suggestions for Authors
This review article provides a comprehensive analysis of the current evidence regarding the use of autologous platelet-rich plasma (PRP) in treating stress urinary incontinence (SUI) in women. The authors highlight PRP’s regenerative potential, its mechanisms of action, and its clinical applications while addressing gaps in standardization, long-term efficacy, and optimal protocols. The review is well-structured and relevant to gynecology and regenerative medicine, offering a balanced perspective on PRP’s promise and limitations.
The review thoroughly examines PRP’s biological basis, preparation methods, and clinical outcomes in SUI, aligning with the integral theory and hammock hypothesis. The authors effectively identify key knowledge gaps, such as the lack of standardized protocols and long-term follow-up data. The discussion on patient-specific determinants (e.g., age, platelet count) and injection-site strategies adds practical value for clinicians.
Nevertheless, while variability in PRP preparation is noted, the review could propose a tentative framework for standardization based on existing evidence (e.g., centrifugation parameters, leukocyte content). The section on durability of PRP effects (Section 9) is limited by sparse data. A meta-analysis of available follow-up studies (even if small) would strengthen conclusions. The contradictory results of single-injection studies (Section 6) warrant deeper discussion. Are outcomes influenced by placebo effects or suboptimal dosing?
The review identifies a clear gap: the need for large-scale RCTs with standardized protocols. However, it could explicitly link this to clinical adoption barriers (e.g., regulatory hurdles, reimbursement). Recent advancements (e.g., PRP combined with lasers or stem cells) are mentioned but not critically evaluated for their potential to address current limitations.
In addition I would like to make some specific comments:
Section 3 (Content of PRP): the classification into P-PRP, LR-PRP, and LP-PRP is useful but could be summarized in a table for clarity.
Line 120: Clarify whether "catabolic effect" of LR-PRP is universally accepted or contested by other studies.
Table 1 (hypothetical): Compare centrifugation protocols (speed, time, platelet yield) from cited studies to highlight optimal parameters.
Line 210: The statement about "hyper-concentrations" being counterproductive needs references to support the claimed threshold (1.0–1.5 × 10⁶/μL).
Line 250: The 60% efficacy rate (Cheng-Yu et al.) should be contextualized with effect sizes (e.g., reduction in ICIQ-SF scores).
Line 280: Discuss why the placebo-controlled trial (Ashton et al.) showed no significant difference. Was the PRP preparation suboptimal?
Figures/Tables: It would be better to add a flowchart summarizing PRP’s mechanism in SUI (e.g., growth factors → tissue repair → urethral support) and include a table comparing efficacy rates, injection protocols, and follow-up durations across key studies.
Thus, the manuscript is well-organized but could benefit from subheadings in longer sections (e.g., "PRP in ISD" under Section 7). The hypotheses (e.g., PRP’s role in ligament repair) are plausible but require more direct evidence from SUI-specific studies. Methods for PRP preparation are described variably. A consolidated protocol recommendation would aid reproducibility. The conclusions are consistent with the evidence but should explicitly call for multicenter RCTs to address standardization and long-term outcomes. However there are some gaps to be filled in: 1. Enhance visual aids (tables/figures) to summarize key data. 2. Deeper critique of placebo effects and protocol heterogeneity. 3. Explicit recommendations for future research (e.g., consensus on centrifugation, injection frequency).
Author Response
- Nevertheless, while variability in PRP preparation is noted, the review could propose a tentative framework for standardization based on existing evidence (e.g., centrifugation parameters, leukocyte content). The section on durability of PRP effects (Section 9) is limited by sparse data. A meta-analysis of available follow-up studies (even if small) would strengthen conclusions. The contradictory results of single-injection studies (Section 6) warrant deeper discussion. Are outcomes influenced by placebo effects or suboptimal dosing?
Response: Thank you for these thoughtful and constructive comments. We fully agree that variability in PRP preparation is an important challenge in the field. However, after careful consideration, we concluded that proposing a standardized framework at this stage would be premature, given the limited number of studies in gynecology that clearly report preparation techniques (including centrifugation speed, duration, platelet/leukocyte concentration, use of activators). The current body of evidence lacks sufficient consistency and detail to support even a tentative protocol recommendation, which we have now explicitly acknowledged in the discussion. Regarding the durability of PRP effects, we agree that this is a key issue. We have clarified that available data on long-term outcomes are sparse, with most studies providing follow-up of only up to 12 months, and very few up to 15 months. As such, we emphasized the need for future studies with extended follow-up periods to assess the true regenerative potential and duration of therapeutic effect.
We have also included a recommendation for a future meta-analysis, once more standardized and transparent data are available, and we note this as an essential next step to strengthen clinical evidence and draw more definitive conclusions.
Finally, in response to your comment on the contradictory results of single-injection studies, we expanded the discussion to more deeply explore the possible influence of placebo effects and suboptimal dosing. We now address how limited protocols may not fully reflect the regenerative potential of PRP, and we emphasize the need for dose-response studies to better define optimal treatment regimens.
- Section 3 (Content of PRP): the classification into P-PRP, LR-PRP, and LP-PRP is useful but could be summarized in a table for clarity.
Response: Thank you for the feedback. In this revised manuscript we have included a table with this classification for better clarity. Please see Table 1.
- Line 120: Clarify whether "catabolic effect" of LR-PRP is universally accepted or contested by other studies.
Response: Thank you for the feedback. In this revised manuscript we have clarified that is not universally accepted the "catabolic effect" of LR-PRP.
- Table 1 (hypothetical): Compare centrifugation protocols (speed, time, platelet yield) from cited studies to highlight optimal parameters.
Response: Thank you for this suggestion. In response, we have created tables (Table 2,3,4), which compares the centrifugation protocols (speed, duration, and reported platelet yield) from the cited studies. This addition aims to highlight the methodological variability and allow readers to identify potential trends toward optimal parameters.
- Line 210: The statement about "hyper-concentrations" being counterproductive needs references to support the claimed threshold (1.0–1.5 × 10⁶/μL).
Response: Thank you for pointing this out. We have now added appropriate references to support the statement regarding the potential counterproductive effects of platelet hyper-concentrations above the 1.0–1.5 × 10⁶/μL threshold.
- Line 250: The 60% efficacy rate (Cheng-Yu et al.) should be contextualized with effect sizes (e.g., reduction in ICIQ-SF scores).
Response: Thank you for your feedback. We have revised that statement.
- Line 280: Discuss why the placebo-controlled trial (Ashton et al.) showed no significant difference. Was the PRP preparation suboptimal?
Response: Thank you for this observation. We have now discussed in the manuscript that the lack of significant difference in the placebo-controlled trial by Ashton et al. may be related to suboptimal or insufficiently reported PRP preparation parameters, such as platelet concentration, leukocyte content, and activation methods. We note that inadequate dosing or reduced biologic activity could have limited the regenerative potential, underscoring the need for transparent and optimized preparation protocols in future studies.
- Figures/Tables: It would be better to add a flowchart summarizing PRP’s mechanism in SUI (e.g., growth factors → tissue repair → urethral support) and include a table comparing efficacy rates, injection protocols, and follow-up durations across key studies.
Response: Thank you for these thoughtful and constructive comments. We added to the revised manuscript a flowchart and tables (1,2,3,4) for better clarification and accuracy.
- Thus, the manuscript is well-organized but could benefit from subheadings in longer sections (e.g., "PRP in ISD" under Section 7). The hypotheses (e.g., PRP’s role in ligament repair) are plausible but require more direct evidence from SUI-specific studies. Methods for PRP preparation are described variably. A consolidated protocol recommendation would aid reproducibility. The conclusions are consistent with the evidence but should explicitly call for multicenter RCTs to address standardization and long-term outcomes. However there are some gaps to be filled in: 1. Enhance visual aids (tables/figures) to summarize key data. 2. Deeper critique of placebo effects and protocol heterogeneity. 3. Explicit recommendations for future research (e.g., consensus on centrifugation, injection frequency).
Response: We appreciate your constructive feedback. In response, we have enhanced the visual aids by adding additional tables and figures summarizing key data, including PRP preparation methods, reported efficacy, and ISD-specific studies. We have also expanded the discussion to provide a deeper critique of placebo effects and protocol heterogeneity, emphasizing how these factors impact the interpretation of outcomes. Furthermore, we have strongly recommended further research focusing on standardized preparation protocols, optimized injection techniques, and robust data collection from SUI-specific studies, particularly to clarify PRP’s role in ligament repair. The narrative review explicitly calls for multicenter randomized controlled trials to establish consensus on centrifugation parameters, injection frequency, and long-term outcomes.

Round 2
Reviewer 1 Report
Comments and Suggestions for Authors
This is an interesting review although very broad.
The search strategy should be also shown on a flowchart, as many good and significant recent papers on PRP topics are not included.
I would not refer reviews and meta-analysis, maybe just the most recent ones for each subtopic, but I would include the largest comparative studies (RCT and not).
Limitations, particularly concerning the different PRP products should be described and discussed with suggestions for further future studies.
Author Response
Response to Reviewer 1
This is an interesting review although very broad.
The search strategy should be also shown on a flowchart, as many good and significant recent papers on PRP topics are not included. I would not refer reviews and meta-analysis, maybe just the most recent ones for each subtopic, but I would include the largest comparative studies (RCT and not).
Response: We thank the reviewer for this valuable suggestion. In the revised manuscript, we have added a detailed flowchart to illustrate our search strategy and study selection process (Figure 1). We have also carefully reviewed the literature to ensure the inclusion of significant recent studies, such as:
- Clavijo J, Boggia B, Castillo E, Gutierrez F, Maglia L, Badia H. Platelet rich plasma peri-urethral injections for the treatment of stress urinary incontinence. CASMU Hospital, Hospital Pereira-Rossell; 2021. ICS 2021, S18, abstract 236.
- Grigoriadis T, Kalantzis C, Zacharakis D, Kathopoulis N, Prodromidou A, Xadzilia S, Athanasiou S. Platelet-rich plasma for the treatment of stress urinary incontinence—a randomized trial. Urogynecology (Phila). 2024;30(1):42-49. doi:10.1097/SPV.0000000000001378.
- Bedir S, Bastawesy M, Heweidy M, Elhamamy N. Autologous platelet-rich plasma versus phytoestrogen gel in management of menopausal women with stress urinary incontinence. Evidence Based Women’s Health J. 2025;15(15):1-7. doi:10.21608/ebwhj.2024.314100.1352.
Limitations, particularly concerning the different PRP products should be described and discussed with suggestions for further future studies.
Response: We thank the reviewer for highlighting this important point. The revised limitations section now explicitly addresses the heterogeneity of PRP products, preparation methods, and reporting standards. We note that:
Future studies should incorporate extended follow-up and standardized outcome measures to better evaluate the durability and clinical utility of PRP in the management of women with SUI. At present, proposing a standardized PRP preparation framework in gynecology is premature, as few studies report sufficient methodological detail to support evidence-based recommendations.

Reviewer 2 Report
Comments and Suggestions for Authors
I confirm that the authors corrected all the mistakes and accepted the recommendations and modified the manuscript according them. Thus, I suppose that the paper could be accepted for the publication.
Author Response
Response to Reviewer 2
I confirm that the authors corrected all the mistakes and accepted the recommendations and modified the manuscript according them. Thus, I suppose that the paper could be accepted for the publication.
Response: We sincerely thank you for your time, effort, and valuable feedback during the review process. Your constructive recommendations have greatly helped us improve the clarity and quality of our manuscript. We are pleased to hear that the revised version meets your expectations, and we truly appreciate your recommendation for acceptance.
